# Bioinformatics and Connectivity Map Analysis Suggest Viral Infection as a Critical Causative Factor of Hashimoto’s Thyroiditis

**DOI:** 10.3390/ijms24021157

**Published:** 2023-01-06

**Authors:** Dong-Woo Lim, Min-Seo Choi, Seok-Mo Kim

**Affiliations:** 1Department of Diagnostics, College of Korean Medicine, Dongguk University, Goyang 10326, Republic of Korea; 2Institute of Korean Medicine, Dongguk University, Goyang 10326, Republic of Korea; 3Gangnam Severance Hospital, Department of Surgery, Yonsei University College of Medicine, Seoul 06273, Republic of Korea

**Keywords:** Hashimoto’s thyroiditis, bioinformatics, viral infection, mRNA splicing, GSEA, autoimmune disease, CMap

## Abstract

Hashimoto’s thyroiditis (HT) is a common autoimmune disease, and its prevalence is rapidly increasing. Both genetic and environmental risk factors contribute to the development of HT. Recently, viral infection has been suggested to act as a trigger of HT by eliciting the host immune response and subsequent autoreactivity. We analyzed the features of HT through bioinformatics analysis so as to identify the markers of HT development. We accessed public microarray data of HT patients from the Gene Expression Omnibus (GEO) and obtained differentially expressed genes (DEGs) under HT. Gene Ontology (GO) and KEGG-pathway-enrichment analyses were performed for functional clustering of our protein–protein interaction (PPI) network. Utilizing ranked gene lists, we performed a Gene Set Enrichment Analysis (GSEA) by using the clusterprofiler R package. By comparing the expression signatures of the huge perturbation database with the queried rank-ordered gene list, a connectivity map (CMap) analysis was performed to screen potential therapeutic targets and agents. The gene expression profile of the HT group was in line with the general characteristics of HT. Biological processes related to the immune response and viral infection pathways were obtained for the upregulated DEGs. The GSEA results revealed activation of autoimmune-disease-related pathways and several viral-infection pathways. Autoimmune-disease and viral-infection pathways were highly interconnected by common genes, while the HLA genes, which are shared by both, were significantly upregulated. The CMap analysis suggested that perturbagens, including SRRM1, NLK, and CCDC92, have the potential to reverse the HT expression profile. Several lines of evidence suggested that viral infection and the host immune response are activated during HT. Viral infection is suspected to act as a key trigger of HT by causing autoimmunity. SRRM1, an alternative splicing factor which responds to viral activity, might serve as potential marker of HT.

## 1. Introduction

Hashimoto’s thyroiditis (HT) is one of the most prevalent autoimmune diseases (AIDs) [1] and a common cause of hypothyroidism in developed countries [2]. The pathologic features of HT include extensive inflammation in thyroid tissue caused by infiltrating CD4+ T lymphocytes [3] and the presence of autoantibodies, causing follicular cell damage and ultimately leading to disrupted thyroid function [4]. The clinical phenotype of HT can vary from asymptomatic to severe symptoms of hypothyroidism, including weight gain, menstrual disorders, and heat intolerance [5]. Although HT is an organ-specific disease, the disease might be related to other AID or systemic autoimmune disorders in many cases [6]. The proposed risk factors for HT include genetic susceptibility, sex (female), and senescence, as well as environmental triggers such as iodine uptake, drugs, chemicals, and viral infections [7]. However, the exact mechanism underlying thyroid autoimmunity in HT remains unclear. 

AIDs lead to tissue injury caused by the autoreactivity of T- and B-cell-mediated responses [8]. Mechanisms of immune tolerance generally prevent autoimmune reactions through the fine-tuning process of positive and negative selection of potentially autoreactive lymphocytes [9]. The failure of these selection mechanisms gives rise to autoreactive T cells or antibodies, which may appear several years prior to the manifestation of AID [10,11]. Genetic susceptibility and environmental factors determine one’s predisposition to AID [12]. 

Genetic predisposition and expression changes influence the autoimmune thyroid disease course by affecting host immunoreactivity or antigen presentation/recognition [12]. Polymorphisms in IL2 and CTLA4 or the upregulation of MHC Class II molecules are implicated in the etiology of AID [10]. MHC, which is also known as human leukocyte antigen (HLA) in humans, plays a key role in AID by helping the immune system distinguish self from foreign, owing to its unlimited allelic diversity [13]. HLA alleles (HLA-DPB1) and their variants (HLA-DPB1*02:02 and HLA-DPB1*05:01) have been described as contributors to the early pathogenesis of autoimmune thyroiditis [14]. However, genetic susceptibility alone is often insufficient to give rise to autoimmunity [15]. 

Recently, viral infection has emerged as an attractive environmental trigger of autoimmunity, with multiple mechanisms described for different AIDs [16]. Molecular mimicry, bystander activation, and epitope spreading are the three major mechanisms underlying virus-induced autoimmunity [17]. Molecular mimicry is based on cross-reactivity due to the structural similarity between pathogen (viral particles) and self (self-antigens), providing a basis for virus-induced autoimmunity [18]. Subsequent tissue damage results in the release of damage-associated molecular patterns (DAMPs) that activate TLRs, leading to amplified immune activation [19]. Likewise, biomolecule-based observation has been indicating viral infections as a key factor in the induction and development of autoimmune diseases. 

A case study reported HT onset after herpes simplex virus infection (3 to 6 months) in three patients, as indicated by the presence of IgM and IgG antibodies against the virus [20]. A study of 42 HT patients revealed a high prevalence of Epstein–Barr virus infection in patient tissue (n = 42) [21]. A larger study showed a clear association of hepatitis C virus (HCV) infection and thyroid autoimmunity [22]. A systematic review of 12 studies compared epidemiological differences in thyroid dysfunction between HCV-infected and non-HCV-infected patients, indicating an increased risk of thyroid dysfunction in the former [23]. 

Recent advances in microarray profiling and bioinformatics tools have enabled the analysis of massive transcriptome data, providing insight into the expression changes observed during various diseases. The Gene Expression Omnibus (GEO) database is a public repository with microarray, next-generation sequencing (NGS), and other genomics data that provides access to large datasets submitted by various researchers [24]. Gene Set Enrichment Analysis (GSEA) is a powerful analytical method used for interpreting gene-expression data via Gene Ontology (GO) terms or other gene-set collections [25]. The Connectivity Map (CMap), a comprehensive, large-scale perturbation database containing 1.5 million gene expression profiles from cultured human cells, can be used to identify potential therapeutic targets or drugs for the submitted gene signature [26]. These bioinformatics tools can be utilized to address various biomedical issues by deciphering information hidden in a large number of biological datasets [27]. 

An investigation into the biomarkers of HT should be conducted to improve diagnosis and provide feasible medical treatment. A previous study on HT utilized the GEO microarray database yet only suggested hub genes deduced from protein–protein interaction (PPI) network of DEGs [28]. In this work, we extracted a list of DEGs from the GEO microarray database and divided them into functional clusters via PPI network construction and performed an overrepresentation analysis (ORA). Based on a ranked gene list, we conducted GSEA to scrutinize the skewed distribution of genes related to specific BP terms and KEGG pathways in the HT group, visualizing these through various R packages. Finally, we identified promising targets and compounds for HT via a CMap analysis.

## 2. Results

### 2.1. Validation of GEO Data

Processed data from the GSE138198 human thyroid tissue microarray dataset was validated by checking the distribution in a boxplot and the heatmap clustering of samples. The boxplot revealed median-centered values, indicating that the data are well-normalized for all samples (Figure 1A). A UMAP plot located each sample in a reduced dimension, using Euclidean distance, concluding that samples within groups are in close proximity (HT vs. TN), and, on the contrary, at long distance between the two groups (Figure 1B). Likewise, the hierarchical correlation heatmap indicated that samples are clustered by groups (Figure 1C). 

### 2.2. Identification of DEGs Extracted from HT

The volcano plot displayed the distribution of DEGs by their fold change and *p*-value, as was distinguishable by color (Figure 1D). After the validation of gene identifiers, a total of 838 upregulated DEGs and 595 downregulated DEGs were confirmed between the HT vs. TN groups. Lists of the 15 most significant up- or downregulated genes are presented in Table 1. KIF5B and PTH were confirmed as the most significantly up- and downregulated DEGs, respectively, in the HT group as compared to the TN group. 

### 2.3. Functional Clustering Analysis of DEGs Reveals the Pathological Characteristics of HT

The full PPI network of up- and downregulated DEGs is presented in Appendix A. A further analysis of functional modules was conducted by using the full PPI network with regard to GO (BP) and KEGG terms (Figure 2 and Figure 3). PPI modules were in line with the general characteristics of HT. Upregulated DEGs were grouped in three notable clusters whose gene lists overrepresented the immune response (viral infection), immune system regulation (autoimmune disease), and RNA translation (ribosome), with scores of 13.467, 8.214, and 6.857, respectively (Figure 2A–C). Meanwhile, downregulated DEGs were grouped into three remarkable clusters enriched in muscle contraction (cardiomyopathy), translation (ribosome), and oxidative phosphorylation (cellular respiration and its related diseases), with scores of 18.842, 11.091, and 7.60, respectively (Figure 3A–C). 

### 2.4. BP GSEA Results Revealed a Strong Connection between HT and Immune Regulation 

The BP GSEA results were presented in a dot plot, which clearly showed the significantly activated or inactivated biological processes in HT (Figure 4A). As expected, immune-response-related BPs were upregulated in HT patients. In contrast, muscular processes were distinctively downregulated in HT patients. Subsequently, the Emapplot showed close interactions between the most significant BP terms based on overlapping genes (Figure 4B). The BP term ‘cytokine production’ had solid interactions with other terms, including ‘positive regulation of immune system process’, ‘regulation of immune response’, ‘hemopoiesis’, and ‘cellular response to cytokine’. These five essential BP terms and enriched genes were further plotted by using the Cnetplot function (Figure 4C). Twenty-seven genes, including CASP8, IL6, TNF, IL1B, and SOCS1, were common between these five BPs, which play key roles in cytokine and inflammatory signaling (Appendix A). The Ridgeplot analysis illustrated that the overall distribution of component genes consists of each BP (Figure 4D). Similarly, BP terms related to various immune responses and cytokine production exhibited increased activity. The term ‘lymphocyte activation’ showed predominant results in regard to both the *p*-value and NES, exhibiting enriched distributions of activated genes (Figure 4E). The top BP GSEA results for the HT group are presented in Table 2. Appendix A presents the top five BPs or pathways and the distributions of associated genes. 

### 2.5. GSEA Results Suggested a Link between Autoimmune Thyroiditis and Viral Infections

The association between HT and other disease pathways was investigated via the KEGG GSEA of HT transcriptional profiles from microarray data. The KEGG GSEA dot plot showed a predominance of viral-infection-related pathways (including human T-cell leukemia virus infection, human cytomegalovirus infection, Epstein–Barr virus infection, herpes simplex virus 1 infection, etc.) in the activated panel, with strong statistical significance (Figure 5A). The Emapplot displayed a number of interconnected viral disease pathways, while the Epstein–Barr virus infection pathway was a hub in the network (Figure 5B). Five representative KEGG pathways with their connected genes were plotted, suggesting that TNF and IL6 were common targets of all five pathways (Figure 5C). The Ridgeplot showed a distribution of gene expression (activated) in several viral-infectious diseases’ pathways, such as herpes simplex virus 1 infection (Figure 5D). Upon a closer look, the comparison of GSEA plots between herpes simplex virus 1 infection and autoimmune thyroid disease revealed that both had a skewed distribution (activated) of genes with a high Normalized Enrichment Score (NES) (Figure 5E). The top KEGG GSEA results of the HT group are presented in Table 3. 

### 2.6. Common DEGs Involved in Two Intuitive Pathways Provide Insight into Disease Etiology 

As our results repeatedly indicated a high correlation of HT with viral infectious disease, we mapped both pathways and colored DEGs (red to green) according to their relative expression levels, using the PathView R package (Figure 6A,B). Common genes between the two pathways were plotted by using the Cnetplot function with modified arguments to investigate only pathways of interest. Of note, two seemingly independent pathways shared many upregulated MHC Class I and II genes (Figure 7). These 14 HLA genes consisted of 2 MHC Class I and 12 MHC Class II molecules. Due to the unillustrated intracellular signaling pathway in the autoimmune thyroid disease KEGG map, other immune-related intersecting genes were not presented as results (Figure 6B). 

### 2.7. CMap Analysis Revealed Potential Markers and Drugs Based on the HT Gene Signature

The CLUE webtool deduced potential target genes and compounds for HT by comparing the gene signature against a gene-expression database based on perturbations. Arranged by their median CMAP score, 19 KD gene perturbations were assumed to restore the gene-expression profiles of the queried signature (HT), with CMap-score criteria below −90 (Figure 8A). Six compound perturbations were expected to reverse the gene signature with the same CMap score as criteria (Figure 8B). Among the 19 KD gene perturbagen lists, three genes (SRRM1, NLK, and CCDC92) actually exhibited a significantly increased expression (*p* < 0.05) in the HT group (Figure 8C). The SRRM1 expression fold change, *p*-value, and transcriptional activity score (TAS) were most prominent among the three genes. 

## 3. Discussion

A US population-based review study reported that AID prevalence is 5–7% and is gradually increasing [10,29]. It has been suggested that both genetic susceptibility and environmental factors act as triggers for the breakdown of tolerance and progress of the disease [6]. Genetic predisposition, usually associated with HLA Class II alleles and other immune mediators, is not sufficient on its own for the increasing prevalence of AIDs [30]; rather, environmental factors, including viral infections, are regarded as major contributors to the incidence of AID [21]. 

Accumulating evidence indicates such a connection between viral infections and the development of AIDs, with an example being the increased prevalence of AIDs following an influenza pandemic reported in a population-based observational study [16,31]. A recent review on SARS-CoV-2 suggested that this virus could also trigger autoimmune responses through molecular mimicry, working in similar ways with other viruses [32]. The most prevalent autoimmune diseases/conditions involved with COVID-19 infection are Guillain–Barre syndrome, immune thrombocytopenic purpura, Kawasaki disease and autoimmune thyroid disease [33]. Investigations on the involvement of SARS-CoV-2 infection in autoimmune thyroid disease development are currently underway [34]. A recent systematic review revealed general characteristics of COVID-19-induced subacute thyroiditis patients [35]. The authors found that subacute thyroiditis to be the most common clinical thyroidal syndrome associated with COVID-19, thus indicating the direct effect of viral infection on autoimmune disease [35]. In another systematic review, the authors pointed out that multifaceted effects of SARS-CoV-2 infection on thyroid functions are variable (thyrotoxicosis, hypothyroidism, and non-thyroidal illness syndrome) and difficult to predict [36]. In this situation, it is of great importance to identify biomarkers to elucidate the link between viral infections and the development of HT.

Our analysis of microarray data revealed characteristic features of HT. The functional cluster analysis of upregulated DEGs provided strong evidence of an association between viral infection and the pathophysiological traits of AID through the activation of innate immune responses (Figure 2) [37]. The GSEA results indicated immune activation in the HT group, with an upregulation of genes implicated in lymphocyte activation, regulatory immune responses, cytokine production, and other related pathways (Figure 4 and Figure 5). Decisively, the GSEA of the herpes simplex virus 1 infection and autoimmune thyroid disease pathways showed similar plots and presented high-ranked significance (each 1st and 44th rank), with the KEGG analysis of DEGs demonstrating an intimate relationship between the two pathways in HT (Figure 5E and Figure 6A,B). 

In our data, of all HLA Class I and II molecules investigated, the expression of 14 genes was significantly upregulated (Figure 7). Aberrant HLA II expression in non-antigen presenting cells, such as epithelial cells and cultured thyrocytes, has been described in a number of AIDs [38]. Similarly, overexpression of HLA Class I during the antiviral response was observed in tissue obtained via core needle biopsy from 46 HT patients [39]. Therefore, the increased expression of both HLA classes might be a general indicator of HT induced by the antiviral immune response. 

Interestingly, ribosome-enriched clusters were noted among both up- or downregulated DEGs (Figure 3). As cellular machinery that is hijacked by viruses, host ribosome factors, including ribosomal proteins (RPLs), play critical roles during viral infection [40]. While detailed functions of the various RPLs are under investigation, these are expected to impact viral replication and gene expression through interactions with viral proteins [40]. The result leads to the speculation of mixed reaction from host cellular defense mechanisms and viral activity. In line with our data, Wu et al. reported similar results from their analysis of blood samples from patients with systemic lupus erythematosus, another AID [41]. When they analyzed the PPI network consisting of DEGs, several RPL genes were implicated as key genes in a module, and the authors explained it as being the outcome of the host immune response to viral infection [41].

SRRM1 is an RNA-binding protein and splicing factor participating in the alternative RNA splicing process [42]. It was suggested that host SRRM proteins are modulated by HIV-1 to facilitate its replication and release [43]. Furthermore, the observed changes in alternative splicing could be either a direct consequence of viral manipulation, the innate immune response, or cellular damage [44]. Dysregulation of post-transcription processes during the antiviral immune response, which is modulated by Type I and III interferons, may lead to autoimmunity [45]. In our study, SRRM1 was suggested as a critical marker of HT, as indicated by the strictly negative CMap score (−99.08) and upregulated expression (log_2_FC = 1.424) in the HT group (Figure 8C). 

While the clinical phenotype of HT differs per case and is difficult to predict, it progresses slowly over months to years [46]. It is highly suggested that patients with autoimmune thyroid disorders should be monitored for the thyroid functions in consideration of the relationship with other systemic autoimmune diseases as well [6]. The predictive Thyroid Events Amsterdam (THEA) score takes into account the TSH, TPOAb levels, and familial background information to estimate the 5-year risk of overt hypothyroidism [47]. Based on our current findings, the evaluation of viral activity, as a key environmental factor in the context of HT, may improve prediction of further development of the disease. To this end, future studies should elucidate the relationship between SRRM1 (or the other alternative splicing factors) and the widely employed clinical HT biomarkers. 

## 4. Materials and Methods

### 4.1. Data Resources and Processing

The human microarray dataset of GSE138198 [48] was accessed via the GEO database (https://www.ncbi.nlm.nih.gov/geo/) at the National Center of Biotechnology Information (NCBI) [24]. The GSE138198 dataset comprises a total of 36 human thyroid tissue microarray samples, including 13 HT tissue samples and 3 normal thyroid (TN) tissue samples. The microarray dataset is based on the GPL6244 (HuGene-1_0-st) Affymetrix Human Gene 1.0 ST Array (transcript (gene) version). 

### 4.2. Screening of DEGs

GEO2R, an interactive web tool for the analysis of GEO datasets, was used to screen DEGs between HT and TN (http://www.ncbi.nlm.nih.gov/geo/geo2r/). GEO2R identified DEGs via GEO queries and the Limma R package from R/bioconductor [49,50]. Genes with an adjusted *p*-value < 0.05 and |log_2_(Fold Change)| > 1 were considered DEGs. Adjustment of the *p*-value was performed via the Benjamini and Hochberg method to reduce false discovery rate. DEGs were visualized with a color-differentiated (by significance and fold change) and labeled volcano plot, using ggplot R packages. A clustered correlation heatmap of all samples was created by using the pheatmap package. 

### 4.3. Construction of PPI Network and Functional Subcluster Analysis 

We used the Search Tool for the Retrieval of Interacting Genes (STRING, https://string-db.org/) online database to construct a PPI of up- or downregulated DEGs [51]. PPI networks were further imported to Cytoscape software (version 3.91) [52] for a functional cluster analysis, which groups genes with similar functions, using the MCODE plugin to yield the top 3 clusters by score [53]. Sets of genes from these clusters were separately subjected to an overrepresentation analysis. 

### 4.4. GO and KEGG Pathway Enrichment Analyses 

GO and KEGG-pathway-enrichment analyses of DEGs were conducted by using the Database for Annotation, Visualization and Integrated Discovery (DAVID database, Accessed on 31th July) [54]. Lists of clustered genes were uploaded with the identifier set to ‘official gene symbol’. The data of enriched terms and KEGG pathways were processed and further visualized as a bubble plot created by using the ggplot2 R package [55]. 

### 4.5. GSEA of GO and KEGG Pathway

GSEA of all ranked genes was performed by using the clusterprofiler R package and the genome-wide annotation package (OrgDb) of Bioconductor [56]. The package provides gseGO and gseKEGG functions for GSEA, using GO (biological process, molecular function, and cellular component) and KEGG annotations. The list of significant gene-set annotations was arranged in the order of normalized enriched score (NES). 

Argument parameters used in gseGO and gseKEGG were as follows: nPerm = 10,000, minGSsize = 3, maxGSSize = 800, pvalueCutoff = 0.05, and pAdjustMethod = ‘none’. Redundant GO terms were removed via the simplify function [57]. Visualization of results obtained from ORA and GSEA analyses was performed by using the enrichplot package [58]. 

### 4.6. Clinical HT Biomarker Identification Via Comparison of Gene Signatures in the CMap Database

We selected the top 150 up- and downregulated DEGs of HT groups and submitted these to the Connectivity Map for analysis by querying the list on the CLUE web tool (https://clue.io/query/, version 1.1.1.43). The results present perturbagens, including KD (gene knock down), CP (compounds), OE (gene overexpression), and PCL (perturbagen lists), aligned by the calculated connectivity score (median tau score), which varies from 100 to −100. 

A negative connectivity score indicates an inversed gene-expression signature between the query and perturbagen, thus implying potential as a promising target or drug [59]. Rank-ordered perturbagens with a CMap connectivity score (tau) < −90 were selected and regarded as significant candidates.

## 5. Conclusions

In conclusion, through the use of bioinformatics approaches for the analysis of microarray data, we provided evidence for the association between viral infection and immune activation in HT. We identified SRRM1, an mRNA splicing factor, as a key player in this association. Clinical validation of our current results remains to be performed. The history of viral infection should be seriously taken into account by clinicians during HT diagnosis. The host immune response to viral infection, especially with alternative mRNA splicing, may provide us helpful indicators of HT development and could be harnessed as diagnostic for therapeutic purposes. Further elucidating the relationship between viral infection and HT may improve therapeutic approaches against this disease. 

## Figures and Tables

**Figure 1 ijms-24-01157-f001:**
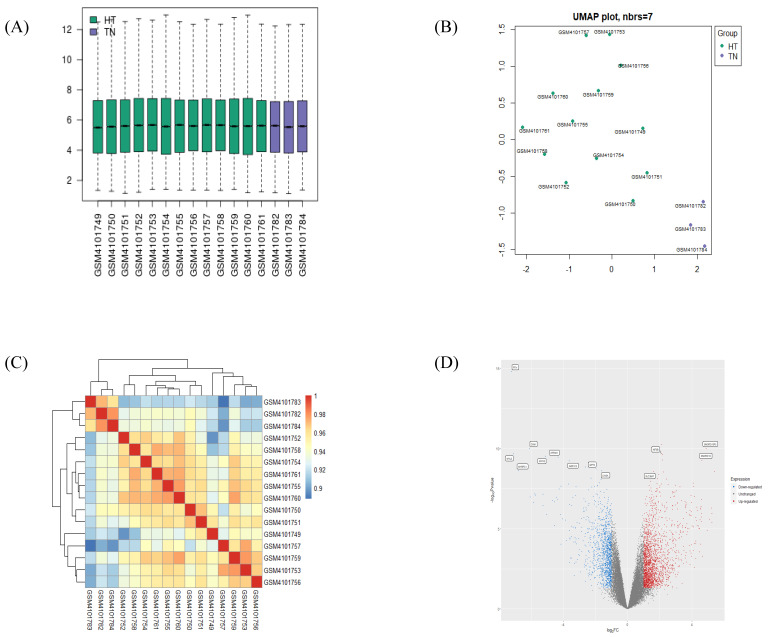
Validation of selected gene-expression profiles from GSE138198. A total of 16 human samples were included, 13 in the HT group and 3 in the TN group. (**A**) Boxplot displays distributions of values of the selected samples from GSE138198 after normalization. (**B**) UMAP plot of 16 normalized samples embedded in a Euclidean space. (**C**) Clustered correlation heatmap showing correlation of microarray profile between 16 samples. (**D**) Volcano plot of significant differentially expressed genes (DEGs) in HT (adjusted *p*-value < 0.05 and |log_2_FC| > 1). Plots were created by using GEO2R or pheatmap R package.

**Figure 2 ijms-24-01157-f002:**
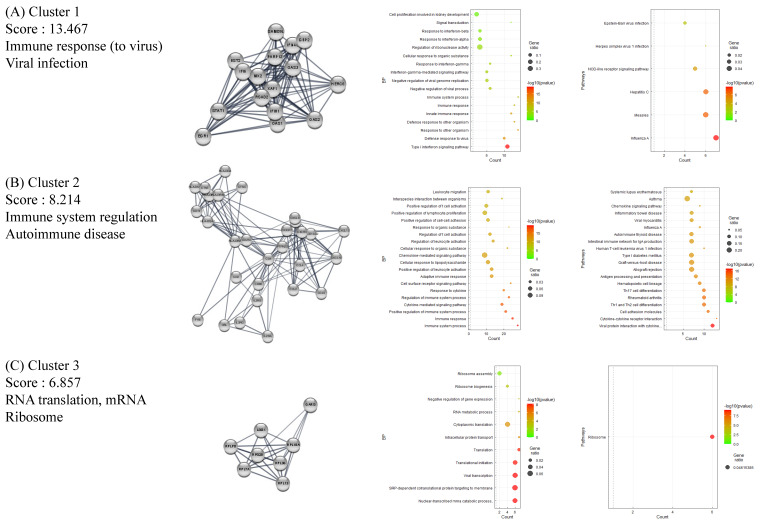
Functional clustering of upregulated DEGs in HT samples, using the MCODE algorithm. Results of BP and KEGG-enrichment analyses presented as dot plots. Each cluster represents a set of highly connected genes. (**A**–**C**) represent Clusters 1–3, respectively, which have the highest clustering scores.

**Figure 3 ijms-24-01157-f003:**
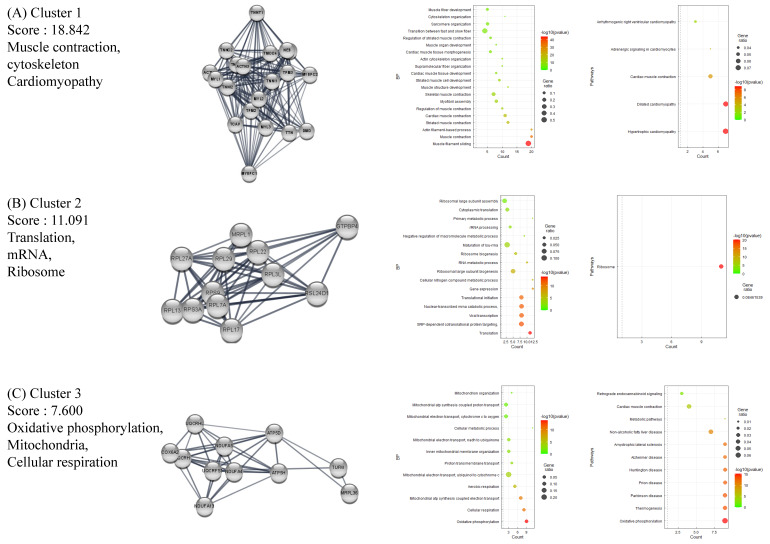
Functional clustering of downregulated DEGs in HT samples, using the MCODE algorithm. Results of BP and KEGG-enrichment analyses presented as dot plots. Each cluster represents a set of highly connected genes. (**A**–**C**) represent Cluster 1–3, respectively, which had the highest clustering scores.

**Figure 4 ijms-24-01157-f004:**
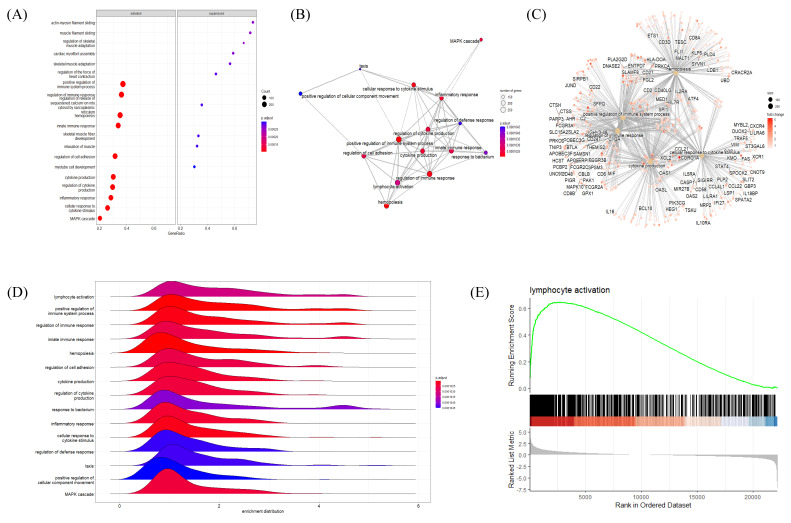
Gene Set Enrichment Analysis (GSEA) plots of BP terms enriched in HT samples. (**A**) Bubble plot of BP terms, (**B**) Emapplot, (**C**) Cnetplot, and (**D**) Ridgeplot with ranked DEGs of the HT group from microarray data. (**E**) GSEA BP plot with ranked DEGs of the HT group from microarray data. All plots were created by using clusterprofiler R package.

**Figure 5 ijms-24-01157-f005:**
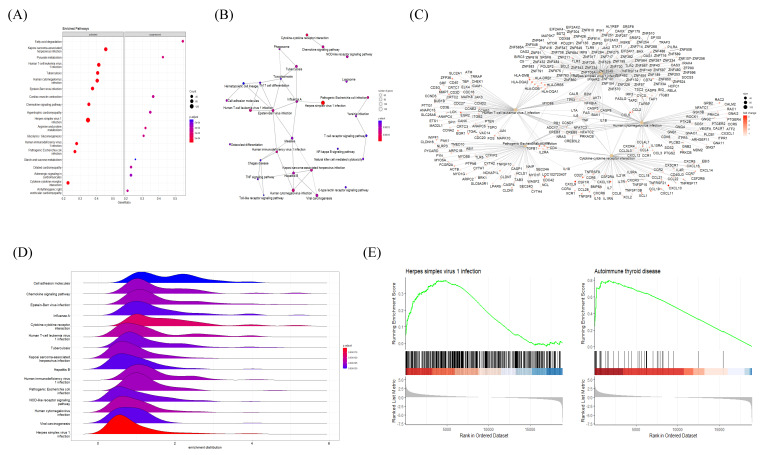
Gene Set Enrichment Analysis (GSEA) plots of KEGG pathways enriched in HT samples. (**A**) Bubble plot of KEGG pathways, (**B**) Emapplot, (**C**) Cnetplot, and (**D**) Ridgeplot analyzed with ranked DEGs of the HT group from microarray data. (**E**) GSEA KEGG plot with ranked DEGs of the HT group from microarray data. All plots were created by using the clusterprofiler R package.

**Figure 6 ijms-24-01157-f006:**
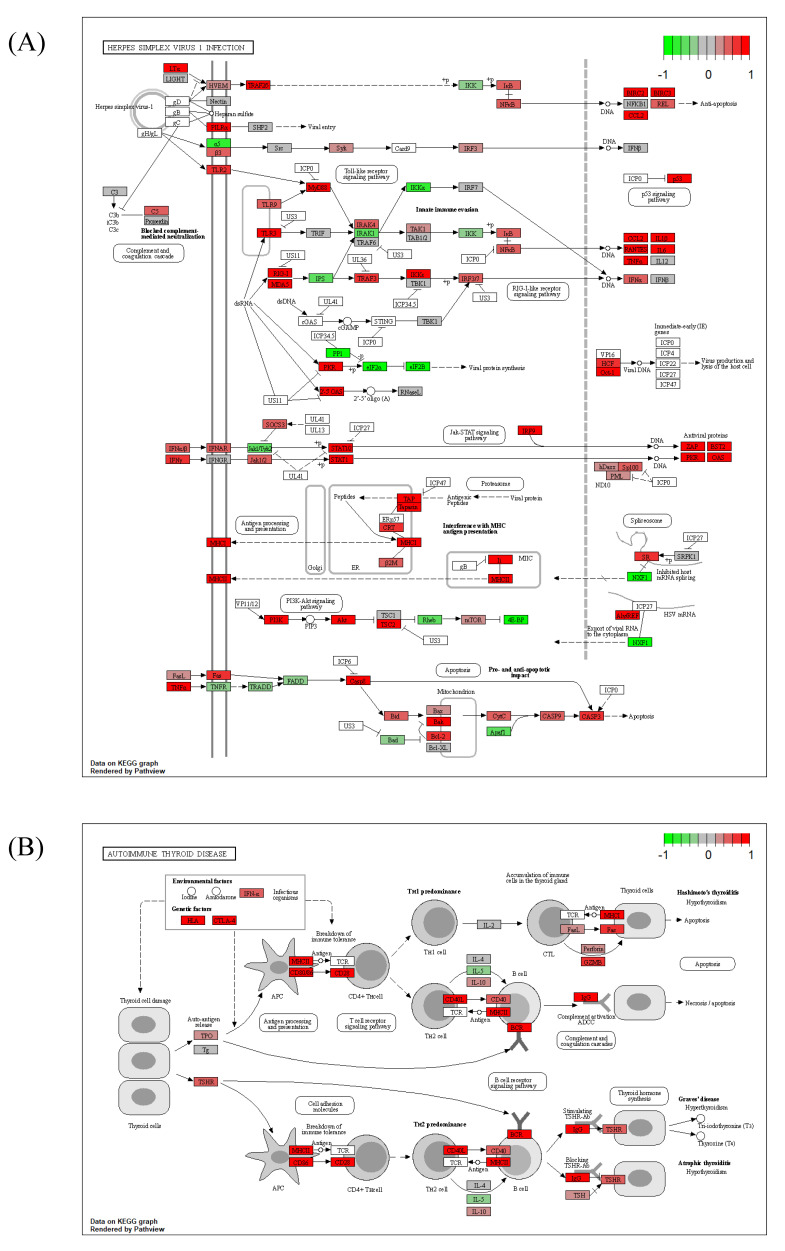
KEGG pathway mapping of (**A**) herpes simplex virus 1 infection pathway and (**B**) autoimmune thyroid disease based on microarray data of the HT group. The plot was created by using PathView R package. The red-to-green color indicates the relative gene expression.

**Figure 7 ijms-24-01157-f007:**
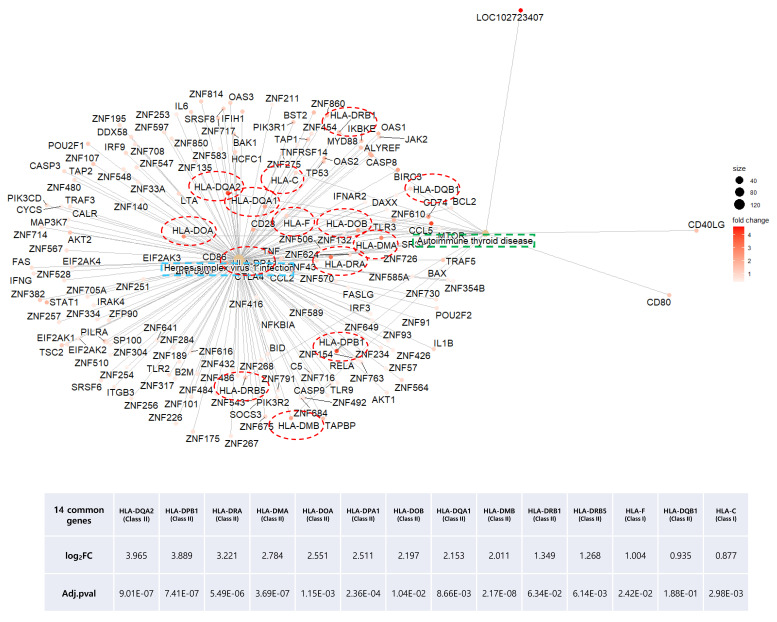
Distribution of overlapping DEGs between the herpes simplex virus infection and autoimmune thyroid disease KEGG pathways of the HT group. Green and blue square boxes indicate pathway names, and the 14 red circles denote overlapping DEGs between the two pathways.

**Figure 8 ijms-24-01157-f008:**
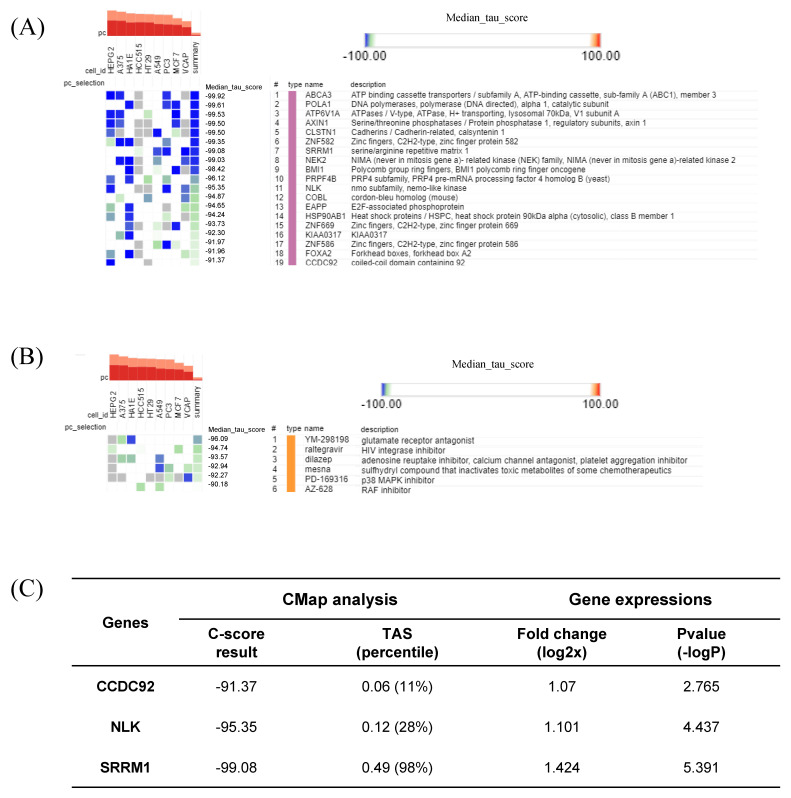
Connectivity Map (CMap) analysis results showing (**A**) significant perturbation genes with the least CMap connectivity score; (**B**) significant perturbation compounds; and (**C**) three significant genes presented with their CMap score, fold-change, *p*-value, and TAS (transcriptional activity score).

**Table 1 ijms-24-01157-t001:** List of top 15 up- or downregulated differentially expressed genes (protein-coding genes only). Genes with |Log_2_(FC)| > 1 and *p*-value < 0.05 were considered to be DEGs.

	Gene Symbol	Gene Name	log_2_(Fold Change)	−Log_10_(P)
Up regulated	*KIF5B*	Kinesin family member 5B	2.092	9.687
	*SLC30A7*	Solute carrier family 30, member 7	1.613	8.584
	*RHNO1*	RAD9-HUS1-RAD1 interacting nuclear orphan 1	1.948	8.386
	*PGK1*	phosphoglycerate kinase 1	1.628	8.321
	*ATP5EP2*	ATP synthase, H+ transporting, mitochondrial F1 complex, epsilon subunit pseudogene 2	1.865	8.107
	*PIGS*	Phosphatidylinositol glycan anchor biosynthesis class S	2.408	8.092
	*CFL1*	Cofilin 1	1.441	7.967
	*ACER3*	Alkaline ceramidase 3	1.833	7.798
	*PARP3*	Poly(ADP-ribose) polymerase family member 3	2.09	7.783
	*ATXN7L1*	Ataxin 7 like 1	2.526	7.756
	*TMA7*	Translation machinery associated 7 homolog	2.999	7.713
	*PTMA*	Prothymosin, alpha	1.341	7.685
	*HLA-DMB*	Major histocompatibility complex, Class II, DM beta	2.011	7.663
	*SAMD9L*	Sterile alpha motif domain containing 9 like	3.262	7.617
	*UNC93B1*	Unc-93 homolog B1 (*C. elegans*)	1.973	7.581
Downregulated	*PTH*	Parathyroid hormone	−7.228	14.79
	*CKM*	Creatine kinase, M-type	−6.094	10.002
	*MYL2*	Myosin light chain 2	−7.113	9.68
	*MYH2*	Myosin heavy chain 2	−5.091	9.511
	*ATP2A1*	ATPase sarcoplasmic/endoplasmic reticulum Ca2+ transporting 1	−4.317	9.467
	*AKR1C3*	Aldo-keto reductase family 1, member C3	−3.636	9.222
	*GPT2*	Glutamic–pyruvic transaminase 2	−1.961	8.731
	*CHGA*	Chromogranin A	−1.615	8.614
	*MYBPC1*	Myosin binding protein C, slow type	−6.775	8.608
	*IGBP1*	Immunoglobulin (CD79A) binding protein 1	−1.63	8.189
	*SAMD8*	Sterile alpha motif domain containing 8	−1.652	8.067
	*SOD1*	Superoxide dismutase 1, soluble	−1.722	7.862
	*AKR1C1*	Aldo-keto reductase family 1, member C1	−3.674	7.796
	*FKBP3*	FK506 binding protein 3	−1.872	7.688
	*TMEM159*	Transmembrane protein 159	−1.87	7.661

**Table 2 ijms-24-01157-t002:** Top 15 BP terms from GSEA of Hashimoto’s thyroiditis samples, ranked by Normalized Enrichment Score (NES).

ID	Description	Enrichment Score	NES	*p*-Value
GO:0002250	Adaptive immune response	0.658	2.637	0.00011
GO:0050870	Positive regulation of T-cell activation	0.674	2.564	0.00012
GO:0046651	Lymphocyte proliferation	0.641	2.502	0.00011
GO:0046649	Lymphocyte activation	0.608	2.494	0.00010
GO:0007159	Leukocyte cell–cell adhesion	0.625	2.483	0.00011
GO:0019724	B-cell mediated immunity	0.684	2.475	0.00013
GO:1990868	Response to chemokine	0.706	2.445	0.00013
GO:0002366	Leukocyte activation involved in immune response	0.619	2.407	0.00011
GO:0002263	Cell activation involved in immune response	0.615	2.389	0.00011
GO:0070661	Leukocyte proliferation	0.606	2.387	0.00011
GO:0050865	Regulation of cell activation	0.584	2.377	0.00011
GO:0002252	Immune effector process	0.580	2.359	0.00011
GO:0002684	Positive regulation of immune system process	0.570	2.347	0.00010
GO:0050776	Regulation of immune response	0.559	2.304	0.00010
GO:0019882	Antigen processing and presentation	0.658	2.286	0.00013

**Table 3 ijms-24-01157-t003:** Top 20 KEGG terms from GSEA of Hashimoto’s thyroiditis samples, ranked by Normalized Enrichment Score (NES).

ID	Description	Enrichment Score	NES	*p*-Value
hsa04061	Viral protein interaction with cytokine and cytokine receptor	0.753	2.683	0.00014
hsa04672	Intestinal immune network for IgA production	0.843	2.672	0.00015
hsa04640	Hematopoietic cell lineage	0.742	2.643	0.00014
hsa05322	Systemic lupus erythematosus	0.785	2.524	0.00015
hsa05320	Autoimmune thyroid disease	0.797	2.499	0.00015
hsa05340	Primary immunodeficiency	0.814	2.480	0.00016
hsa04514	Cell-adhesion molecules	0.647	2.449	0.00013
hsa05330	Allograft rejection	0.821	2.427	0.00016
hsa05310	Asthma	0.837	2.426	0.00016
hsa04658	Th1- and Th2-cell differentiation	0.678	2.405	0.00014
hsa05332	Graft-versus-host disease	0.817	2.385	0.00016
hsa05140	Leishmaniasis	0.693	2.379	0.00014
hsa05150	Staphylococcus aureus infection	0.672	2.365	0.00014
hsa04662	B-cell receptor signaling pathway	0.675	2.361	0.00014
hsa04062	Chemokine signaling pathway	0.607	2.360	0.00013
hsa04940	Type I diabetes mellitus	0.775	2.351	0.00016
hsa05323	Rheumatoid arthritis	0.663	2.341	0.00014
hsa04659	Th17-cell differentiation	0.637	2.313	0.00014
hsa05169	Epstein–Barr virus infection	0.587	2.291	0.00013
hsa04650	Natural-killer-cell-mediated cytotoxicity	0.622	2.274	0.00014

## Data Availability

The datasets generated during and/or analyzed during the current study are available in the Gene Expression Omnibus (GEO) repository, https://www.ncbi.nlm.nih.gov/geo/query/acc.cgi?acc=gse138198.

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
