# Peer review of "Bioinformatics and Connectivity Map Analysis Suggest Viral Infection as a Critical Causative Factor of Hashimoto’s Thyroiditis"

_ijms, 2023, doi:10.3390/ijms24021157_

Round 1
Reviewer 1 Report
The manuscript of Lim and colleagues consider a very innovative and interesting topic. In particular, using bioinformatics methods, the authors found other proofs about the association between specific genetic predispositions and environmental factors, such as viruses, to explain the development of autoimmune disorders, including autoimmune thyroid diseases.
However, the approach of the paper is very specialized and it needs a good level of expertise to understand the methodology and , consequently, the results. This element restricts the access of the manuscript to a limit number of readers.
Said that, I suggest:
-to put the "Methods" part just after introduction and before the results and discussion.
-please consider in the "Introduction" section the following landmark paper: "Autoimmune thyroid disorders" by Antonelli et al, Autoimmunity Review, 2015
- in the "Discussion" section, please expand the part about the association between SARS-COV-2 infection and autoimmune disorders, which is very debating nowadays.
Author Response
Reviewer Comments:
The manuscript of Lim and colleagues consider a very innovative and interesting topic. In particular, using bioinformatics methods, the authors found other proofs about the association between specific genetic predispositions and environmental factors, such as viruses, to explain the development of autoimmune disorders, including autoimmune thyroid diseases.
Author comments:
- First, we would like to express our gratitude on your comments and efforts.
- By using the up-to-date technique of bioinformatics analysis, we tried to explain the most prominent triggers for the onset of autoimmune thyroiditis (Hashimoto’s thyroiditis).
- We have insisted that the genetic factors are not sufficient to explain the onset of autoimmune thyroiditis for the majority of people, due to the various features in onset and progression of the disease.
- Many environmental factors take part in triggering AITD. However, we have found the evidences for viral infection as a major factor among those environmental factors.
However, the approach of the paper is very specialized and it needs a good level of expertise to understand the methodology and , consequently, the results. This element restricts the access of the manuscript to a limit number of readers.
Author comments:
- We fully appreciate that the methodological approaches used in this study are not easy. However, the main outcomes are obvious: viral infection is indicated as the most prominent factor by the transcriptome analysis of thyroid tissue.
- We did our best to explain it as easily as possible in the result / discussion part. We hope readers find it interesting to read and understand our main points.
Said that, I suggest:
-to put the "Methods" part just after introduction and before the results and discussion.
Author comments:
- Thank you for your kind suggestion. We correct it promptly.
-please consider in the "Introduction" section the following landmark paper: "Autoimmune thyroid disorders" by Antonelli et al, Autoimmunity Review, 2015
Author comments:
- We would like to appreciate for recommending us an eminent research paper in this field.
- We added the reference of your suggestion on the manuscript, in page of 1, 15, 16 (referenced as [6]).
- in the "Discussion" section, please expand the part about the association between SARS-COV-2 infection and autoimmune disorders, which is very debating nowadays.
Author comments:
- We are grateful for your comments on the discussion part in relation with SARS-COV-2 infection and autoimmune disorders.
- As many researchers aware, the COVID-19 pandemic affects all areas of human health, evidently with thyroid-related disease too. We have found a lot of studies supporting the connection between COVID-19 infection and onset of autoimmune disease and autoimmune thyroiditis.
- We have reinforced the issue in discussion part (page 15) with appropriate references ([45],[47],[48]) to highlight the importance of outcomes from this study.
- Reference 45 covers observational cause and effect between COVID-19 and several autoimmune diseases.
- Reference 47 is a systematic review that covers characteristics of subacute thyroiditis patient with (after) COVID-19 infection.
- Reference 48 described the variable effects of SARS-CoV-2 infection in thyroid function, thus reminding the importance of the study.
Again, we appreciate your sincere review on our manuscript.

Reviewer 2 Report
Very nice bioinformatics study. Although, as stated by the authors, clinical validation remains to be performed, the results are very interesting and deserve publication.
Author Response
Reviewer Comments:
Very nice bioinformatics study. Although, as stated by the authors, clinical validation remains to be performed, the results are very interesting and deserve publication.
Author commnets:
- Thank you very much for your kind comments on our study. As pointed out, clinical validations of these key markers are needed. We are planning for a study to find evidences for viral infection from thyroid tissue of Korean population with medical history of thyroiditis.
- In future study, we would like to focus on elucidating the direct cause and effect on onset of thyroiditis with provided markers in our manuscript (SRRM1, NLK, CCDC92).
- We appreciate again for your generous opinion.

Round 2
Reviewer 1 Report
Thank you for your answer. In my opinion, now the manuscript is ready to be published